# More Than Meets the Eye:
# Enhancing Multi-Object Tracking Even with Prolonged Occlusions

**Bishoy Galoaa** [1]  **Somaieh Amraee** [1]  **Sarah Ostadabbas** [1]

## Abstract

This paper introduces MOTE (MOre Than meets the Eye), a novel multi-object tracking (MOT) algorithm designed to address the challenges of tracking occluded objects. By integrating deformable detection transformers with a custom disocclusion matrix, MOTE significantly enhances the ability to track objects even when they are temporarily hidden from view. The algorithm leverages optical flow to generate features that are processed through a softmax splatting layer, which aids in the creation of a disocclusion matrix. This matrix plays a crucial role in maintaining track consistency by estimating the motion of occluded objects. MOTE's architecture includes modifications to the enhanced track embedding module (ETEM), which allows it to incorporate these advanced features into the track query layer embeddings. This integration ensures that the model not only tracks visible objects but also accurately predicts the trajectories of occluded ones, much like the human visual system. The proposed method is evaluated on multiple datasets, including MOT17, MOT20, and DanceTrack, where it achieves impressive tracking metrics–82.0 MOTA and 66.3 HOTA on the MOT17 dataset, 81.7 MOTA and 65.8 HOTA on the MOT20 dataset, and 93.2 MOTA and 74.2 HOTA on the Dance-Track dataset. Notably, MOTE excels in reducing identity switches and maintaining consistent tracking in complex real-world scenarios with frequent occlusions, outperforming existing state-of-the-art methods across all tested benchmarks. Code is available at github.com/ostadabbas/MOTE-More-Than-Meets-the-Eye-Tracking.

[1]Augmented Cognition Lab (ACLab), Department of Electrical and Computer Engineering, Northeastern University, Boston, MA, USA. Correspondence to: Sarah Ostadabbas <ostadabbas@ece.neu.edu>.

*Proceedings of the 42$^{nd}$ International Conference on Machine Learning*, Vancouver, Canada. PMLR 267, 2025. Copyright 2025 by the author(s).

## 1. Introduction

Multi-object tracking (MOT) presents a significant challenge in computer vision, with wide-ranging applications such as surveillance (Ahmed et al., 2021; Amraee et al., 2024; Vennila & Balamurugan, 2023), autonomous driving (Gragnaniello et al., 2023), and robotics (Zaeni et al., 2018). The core problem in MOT involves the consistent identification and tracking of multiple objects across successive frames. This task becomes particularly difficult in scenarios where objects are occluded, either partially or fully, making it challenging to maintain accurate tracking (Ciaparrone et al., 2020; Zhang et al., 2021).

The human visual system is remarkably proficient at handling occlusion, effectively estimating the motion of hidden objects by using contextual information from visible surroundings (Saleh et al., 2020). This innate ability to infer the presence and trajectory of objects, even when they are out of view, inspired our approach to MOT. Unlike traditional MOT methods that heavily rely on the visual appearance of objects and often struggle with occlusion, we seek to replicate this perceptual skill through advanced computer vision techniques.

In this work, we introduce MOTE (MOre Than meets the Eye), which fundamentally advances occlusion handling in multi-object tracking through three key innovations: (1) a novel integration of optical flow with transformer-based tracking that enables robust motion prediction during occlusions, (2) an adaptive softmax splatting mechanism that intelligently weights feature propagation based on occlusion patterns, and (3) an enhanced track embedding module (ETEM) that maintains object identity through occlusions by fusing motion and appearance cues. Unlike existing methods like other end to end methods that primarily rely on appearance matching, MOTE explicitly models occlusion dynamics through these complementary components.

MOTE is designed to operate end-to-end as shown in Figure 1. It begins by employing deformable transformers to process input frames and generate pyramid feature maps, which capture detailed spatial information at various scales, providing a comprehensive understanding of the scene. This spatial analysis is followed by optical flow estimation, which

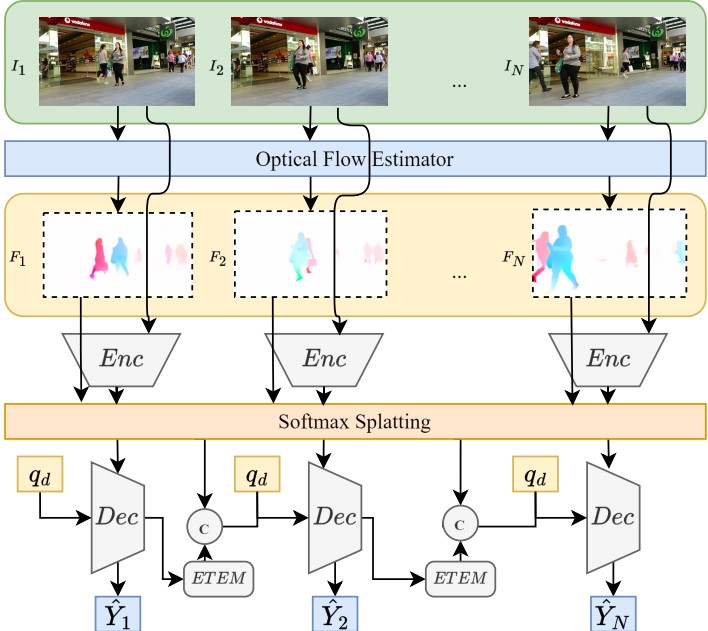

Figure 1. Overview of the MOTE framework: Our method processes video frames $(I_1, I_2, \ldots, I_N)$ to produce robust object tracks $(\hat{Y}_1, \hat{Y}_2, \ldots, \hat{Y}_N)$. An optical flow estimator computes flow fields between consecutive frames, while a deformable DETR encoder (Enc) extracts features from each frame. These elements feed into the softmax splatting module. The output from softmax splatting is then concatenated (denoted by the C symbol) with the output from the Enhanced Track Embedding Module (ETEM), which integrates splatted features into track query embeddings. This concatenated result is processed by a decoder (Dec) with query-driven attention $(q_d)$ to generate the final object tracks. This sophisticated pipeline enables consistent tracking even in scenarios with frequent occlusions and complex scene dynamics.

tracks the movement of objects between consecutive frames–a crucial capability in scenarios where objects might be partially or completely occluded.

A key innovation in our approach is the incorporation of a softmax splatting layer, which generates disocclusion features by merging the pyramid feature maps with the optical flow. This process forms a disocclusion matrix that highlights areas of occlusion and estimates the motion of obscured objects, significantly enhancing the system's tracking capabilities (Niklaus & Liu, 2020). These disocclusion features are then integrated into an enhanced track embedding module (ETEM), which aids in sequence and occlusion estimation.

The significance of MOTE lies in its ability to maintain accurate tracking even under prolonged occlusions, a scenario that challenges most existing MOT systems. By leveraging contextual information and motion estimation, MOTE can infer object trajectories in a manner akin to human visual perception, marking a substantial leap forward in robust multi-object tracking for real-world applications. Our contributions in this paper are as follows:

- We translate the human ability to perceive and track occluded objects into a novel end-to-end multi-object tracking framework, integrating deformable transform-

ers with optical flow estimation to enhance tracking capabilities in occlusion scenarios (see Figure 1).

- We develop a unique softmax splatting layer to generate disocclusion features, which are integrated into the enhanced track embedding module (ETEM) to effectively handle occluded objects and improve tracking accuracy (see Figure 2).

- We conduct extensive ablation studies and evaluations on multiple public datasets (MOT17 (Milan et al., 2016), MOT20 (Dendorfer et al., 2020), Dance-Track(Sun et al., 2022)), demonstrating the effectiveness of our approach in various challenging scenarios, including low frame rates and large camera motions.

- We achieve state-of-the-art performance on standard MOT metrics while significantly improving tracking accuracy during occlusion events, with a 2.4% reduction in identity switches compared to existing methods.

## 2. Related Works

In this section, we summarize key developments in multi-object tracking (MOT) that specifically address occlusion challenges, focusing on approaches that inform our MOTE framework. We examine how different methods handle

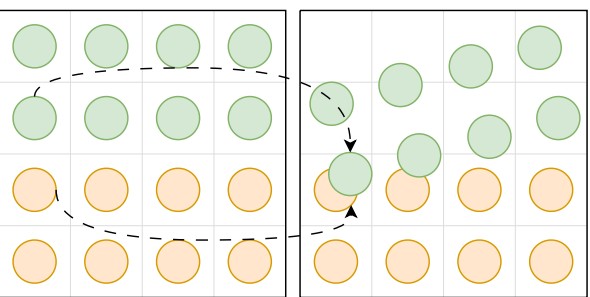 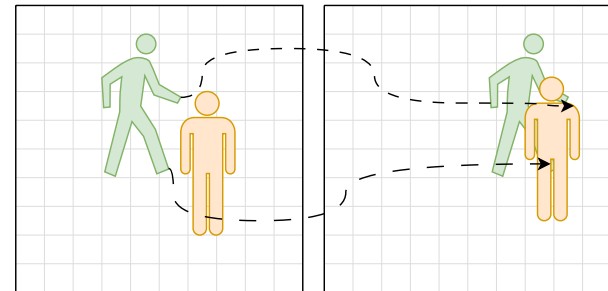

*Figure 2.* Illustration of normal splatting (Left): The orange pixels remain static while the green pixels move down in a shearing manner. Splatting allows scaling of the transform. Application of splatting in an occlusion scenario (Right): Using softmax splatting, the occlusion handling can be effectively translated by ensuring that the motion and visibility of objects are accurately managed, improving multi-object tracking under occlusion conditions.

tracking through occlusions and analyze their limitations, which motivate our work.

### 2.1. CNN-based Methods: Tracking by Detection

CNN-based MOT methods have evolved to better handle occlusion scenarios. While SORT (Bewley et al., 2016) established foundational tracking-by-detection principles, more recent approaches like ByteTrack (Zhang et al., 2022b) specifically target missed detections during occlusions. BoostTrack (Zhang et al., 2023a) advances this paradigm by introducing an innovative detection-tracklet confidence score and enhanced similarity measures including Mahalanobis distance and shape similarity, achieving state-of-the-art performance while maintaining real-time execution. StrongSORT (Du et al., 2023) further refines the process through enhanced trajectory association during partial occlusions. However, these methods still struggle with prolonged occlusions due to their heavy reliance on frame-by-frame detections and limited temporal modeling capabilities. Our work addresses these limitations through advanced occlusion handling and enhanced temporal feature integration.

### 2.2. Transformer-based Methods: End-to-End Frameworks

Recent transformer-based approaches have shown promise in handling occlusions through unified detection and tracking frameworks. MOTR (Zeng et al., 2022) introduces track queries for modeling object persistence across video sequences, enabling end-to-end learning. MOTRv2 (Zhang et al., 2023b) enhances this with bootstrapped pretraining and improved temporal attention. DragonTrack (Galoaa et al., 2025) advances this paradigm by integrating graph convolutional networks with transformer features for enhanced re-identification in complex scenarios. Despite these advances, performance in high-occlusion scenarios remains challenging, particularly when objects disappear for ex-

tended periods. Our MOTE framework builds upon these methods while introducing specific components for robust occlusion management, addressing limitations in handling prolonged occlusions.

### 2.3. Optical Flow-based Trackers

Optical flow estimation has proven crucial for understanding object motion during occlusions. TransMOT (Zhu et al., 2021) demonstrates how flow information can enhance object association in transformer-based systems, while DualFlow (Zhang et al., 2022a) specifically targets occlusions in crowded scenes. These approaches inform our integration of optical flow for maintaining track consistency through occlusion events.

### 2.4. Softmax Splatting and Occlusion Handling in MOT

As illustrated in Figure 2, softmax splatting (Niklaus & Liu, 2020) offers a powerful approach for handling occlusions. Originally developed for video interpolation, this technique effectively reconstructs occluded regions through feature weighting across multiple pyramid levels (Reda et al., 2021). While recent point tracking methods like PointOdyssey (Zheng et al., 2023) and TAPIR (Doersch et al., 2023) have shown success in handling occlusions at a fine-grained level, our work adapts these insights to object-level tracking. By combining transformer architectures with optical flow estimation and softmax splatting (Figure 3), MOTE maintains consistent tracking through extended occlusion periods.

## 3. Introducing MOTE

The proposed MOTE framework, illustrated in Figure 1, processes a sequence of video frames $(I_1, I_2, \ldots, I_N)$ to generate robust object tracks $(\hat{Y}_1, \hat{Y}_2, \ldots, \hat{Y}_N)$. This system integrates advanced computer vision techniques with novel architectural components to handle occlusions effec-

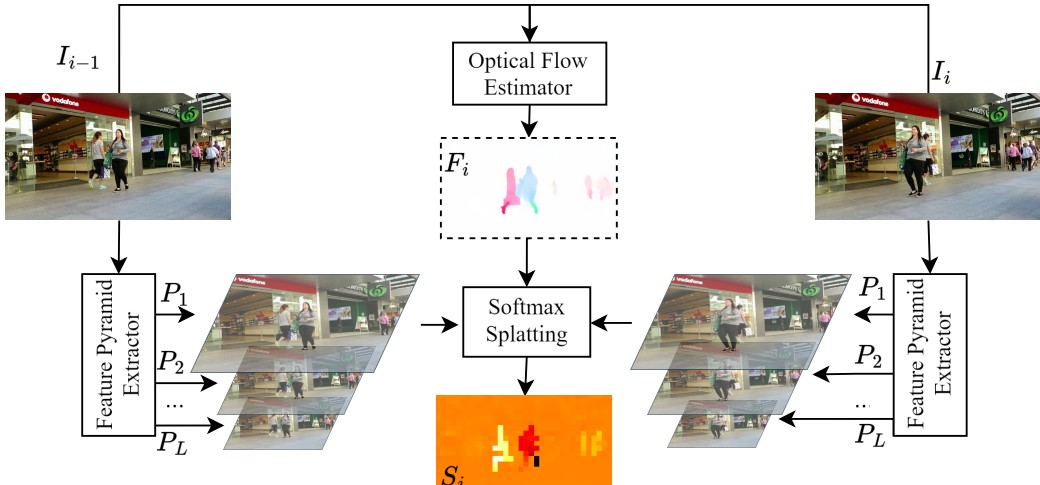

*Figure 3.* Diagram illustrating the softmax splatting method used in the MOTE framework. Frames $I_i$ and $I_{i-1}$ are passed through an optical flow estimator to compute the flow field $F_i$. The feature pyramid extractor within the deformable DETR extracts multi-scale features $P_1$, $P_2$,..., $P_L$ from both frames. The softmax splatting layer then combines the optical flow and the pyramid features to generate the disocclusion matrix $S_i$. In the disocclusion matrix, hotter (darker) areas indicate that the subject is disoccluded and more likely to occlude other objects (less depth), while lighter areas indicate that the object will be occluded (more depth).

tively and maintain consistent object identities throughout complex sequences.

### 3.1. Optical Flow Estimation

The core of our tracking framework is the motion estimation between consecutive frames. We utilize optical flow over conventional Kalman filtering as it provides richer motion cues through pixel-level movement estimation, enabling better handling of occlusions and non-linear trajectories. Specifically, we employ the state-of-the-art RAFT (recurrent all-pairs field transforms) model (Teed & Deng, 2020). For any pair of input frames $I_i$ and $I_{i-1}$, the optical flow $F_{i,i-1}$ is computed as follows:

$$F_{i,i-1} = \text{RAFT}(I_i, I_{i-1}). \tag{1}$$

This step captures inter-frame motion, providing essential information for tracking objects through occlusions and dynamic scene changes.

### 3.2. Feature Extraction and Softmax Splatting

Building on the motion information, our approach employs a deformable DETR encoder to extract multi-scale features from input frames. Softmax splatting, originally designed for video frame interpolation, serves a crucial role in our framework by adaptively combining feature information across different scales with motion cues. The softmax splatting layer, depicted in Figure 3, intelligently merges feature maps $\{P_1^t, P_2^t, \ldots, P_L^t\}$ at time step $t$ from the deformable DETR with optical flow $F$ to maintain object consistency

during occlusions. As visualized in the bottom row of Figure 3, the resulting disocclusion matrix $S_i$ effectively captures occlusion states: darker regions indicate subjects that are disoccluded and more likely to occlude others (less depth), while lighter regions signify subjects that are more likely to be occluded (more depth). This visualization empirically validates our approach's ability to identify and handle occlusion relationships. The normalized weights for each feature level are calculated as:

$$w = \text{softmax}(\theta), \tag{2}$$

where $\theta \in \mathbb{R}^{L+1}$ is a learnable parameter vector that determines importance weights for both feature pyramid levels ($L = 4$ in our implementation) and the optical flow features (accounting for the $+1$ dimension). The softmax operation ensures that features from different scales contribute proportionally to their relevance for tracking. All feature maps are resized to maintain spatial consistency:

$$P_L^{\text{resized}} = \text{interp}(P_L^t, \text{size} = (H_{\min}, W_{\min})), \tag{3}$$

where $L$ represents the number of feature pyramid levels (typically set to 4 in our implementation), and $L + 1$ in eq.5 refers to the additional channel dimension introduced by the optical flow features, enabling dynamic balancing between appearance and motion cues. The optical flow information is processed to match the feature dimensions:

$$F^{\text{expanded}} = F^{\text{resized}} \otimes 1_{C/2}. \tag{4}$$

The final splatted feature $S$ combines these elements

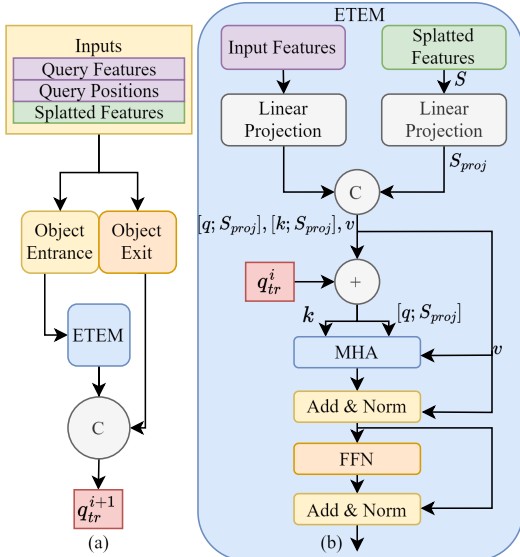

*Figure 4.* Illustration of the ETEM module: (a) the track handling module manages object entries and exits while processing query features, query positions, and splatted features to produce updated track embeddings $q_{tr}^{i+1}$. (b) the enhanced track embedding module (ETEM) processes inputs by applying linear projections to both input and splatted features, followed by concatenation and multi-head attention (MHA). the resulting embeddings are refined through add & norm layers and a feed-forward network (FFN), ultimately updating the track queries.

through a weighted sum:

$$S = \sum_{l=1}^{L} w_l \cdot P_L^{\text{resized}} + w_{L+1} \cdot F^{\text{expanded}}. \quad (5)$$

This mechanism allows our model to dynamically adjust the contribution of each feature level and motion information based on the tracking context, particularly beneficial when dealing with occluded objects where certain feature levels may become more informative than others.

### 3.3. Enhanced Track Embedding Module (ETEM)

The Enhanced Track Embedding Module (ETEM) addresses a critical challenge in occlusion-aware tracking: seamlessly integrating temporal motion information with spatial features while maintaining object identity. Figure 4 illustrates the complete data flow through ETEM. The module processes three inputs: (1) query features $q_{\text{feat}}$ from the transformer encoder, (2) positional encodings $q_{\text{pos}}$, and (3) splatted features $S$ from our softmax splatting layer. These inputs flow through parallel linear projections before concatenation, enabling the model to learn optimal feature combinations for occlusion handling. The concatenated features then undergo multi-head attention with track queries $q_{tr}^i$, followed by feed-forward networks with residual connections,

producing updated track embeddings $q_{tr}^{i+1}$ that maintain consistent object identities through occlusions.

The ETEM, illustrated in Figure 4, consists of two main components: a track handling module and the core ETEM processing block. As shown in Figure 4(a), the track handling module manages object entries and exits while processing three types of inputs: query features, query positions, and splatted features. In Figure 4(b), these inputs undergo parallel linear projections. The input features (query features and positions) are processed through one branch, while splatted features $S$ are processed through another to produce $S_{\text{proj}}$:

$$S_{\text{proj}} = W_S S + b_S, \quad (6)$$

where $W_S$ and $b_S$ are learnable parameters.

These projected features are then concatenated as shown by the 'C' symbol in the figure:

$$[q; S_{\text{proj}}] = [q_{\text{pos}}; q_{\text{feat}}; S_{\text{proj}}], \quad (7)$$

where $[;]$ denotes concatenation. This concatenated representation, along with the current track query $q_{tr}^i$, feeds into the attention mechanism. Following the diagram, for each attention head $i$:

$$\begin{aligned}
Q_i &= q_{tr}^i W_i^Q, \\
K_i &= [q; S_{\text{proj}}] W_i^K, \\
V_i &= [q; S_{\text{proj}}] W_i^V,
\end{aligned} \quad (8)$$

where $W_i^Q$, $W_i^K$, and $W_i^V \in \mathbb{R}^{d \times d_k}$ are learnable matrices and $d_k = 256$. Each attention head computes:

$$\text{head}_i = \text{softmax}\left(\frac{Q_i K_i^T}{\sqrt{d_k}}\right) V_i. \quad (9)$$

The outputs from all eight attention heads are combined and processed through add & norm layers as shown in Figure 4(b):

$$Q' = \text{Concat}(\text{head}_1, \dots, \text{head}_8) W^O, \quad (10)$$

followed by a feed-forward network with residual connections:

$$q_{tr}^{i+1} = \text{FFN}(Q') = \max(0, Q' W_1 + b_1) W_2 + b_2. \quad (11)$$

This architecture, with its carefully designed attention mechanism and residual connections, enables effective processing of both appearance and motion information, maintaining robust tracking even through occlusion events. The sequential processing through MHA, add & norm layers, and FFN, as depicted in Figure 4(b), ensures that the final track embeddings $q_{tr}^{i+1}$ capture rich spatio-temporal relationships for reliable tracking.

## 3.4. Decoder and Object Tracking

The final stage of our pipeline in Figure 1 involves a decoder (Dec) equipped with query-driven attention ($q_d$). This decoder processes the enhanced features produced by the ETEM to generate the final object tracks. By leveraging the rich representation created by the previous modules, we maintain consistent object identities even in the most challenging scenarios.

## 3.5. Loss Function

Our loss function adaptively weights different components based on occlusion states. For each tracked object, we compute an occlusion mask:

$$M_{ij} = \max_{k \neq i} \text{IoU}(b_i, b_k), \tag{12}$$

where $M_{ij}$ measures the maximum overlap between object $i$'s bounding box $b_i$ and any other object $k$'s box $b_k$ in frame $j$. This mask automatically adapts the tracking strategy: high values ($M_{ij} \approx 1$) for heavily occluded objects emphasize temporal and contextual cues, while low values ($M_{ij} \approx 0$) favor appearance-based tracking.

The total loss combines three components weighted by this occlusion mask:

$$\mathcal{L}_{\text{total}} = \mathbb{E}\left[M \odot (\lambda_1 \mathcal{L}_{\text{bbox}} + \lambda_2 \mathcal{L}_{\text{giou}} + \lambda_3 \mathcal{L}_{\text{cls}})\right], \tag{13}$$

where each component serves a specific purpose. The bounding box loss measures coordinate accuracy:

$$\mathcal{L}_{\text{bbox}} = \|b_{\text{pred}} - b_{\text{gt}}\|_1, \tag{14}$$

while the GIoU loss captures overall spatial overlap:

$$\mathcal{L}_{\text{giou}} = 1 - \text{GIoU}(b_{\text{pred}}, b_{\text{gt}}). \tag{15}$$

For classification, we employ focal loss to address class imbalance:

$$\mathcal{L}_{\text{cls}} = -\alpha_t (1 - p_t)^\gamma \log(p_t), \tag{16}$$

where $p_t$ is the estimated probability for the target class, $\alpha_t$ weights different classes, and $\gamma$ controls the focus on hard examples. The effectiveness of this occlusion-aware weighting scheme is shown in Sec. 4.6 ablation studies.

# 4. Experimental Results

In this section, we present the experimental results that demonstrate the effectiveness of our proposed MOTE framework. We begin by detailing the implementation setup, including hardware specifications, datasets, and evaluation metrics. Following this, we provide a comprehensive comparison of MOTE against state-of-the-art methods, highlighting its superior performance across multiple challenging datasets. We also offer a qualitative analysis to showcase

MOTE's robustness in handling occlusions. Finally, we conduct an ablation study to investigate the contributions of different components within MOTE, providing insights into the factors that drive its enhanced tracking capabilities.

## 4.1. Implementation Details

**Model Configuration:** The ETEM module uses a feature dimension of 256 throughout all layers, with 8 attention heads in the multi-head attention mechanism. The softmax splatting module processes feature pyramids with scales [1/32, 1/16, 1/8, 1/4] of the input resolution. For optical flow estimation, we use RAFT with 20 iterative refinement steps, as our ablation studies show this provides optimal balance between accuracy and computational efficiency.

**Training Protocol:** The model is trained end-to-end using AdamW optimizer with an initial learning rate of 1e-4 and cosine decay schedule. We employ standard data augmentation techniques including random horizontal flipping and random cropping. All loss components are weighted as $\lambda_1 = 5.0$, $\lambda_2 = 2.0$, and $\lambda_3 = 2.0$ based on validation performance.

**Setting:** The experiments were conducted on a system equipped with an Intel Xeon CPU E5 2.40GHz, 4 A100 GPUs, and 16 GB of RAM. This setup provided the necessary computational power to handle the intensive training and evaluation processes. The model was trained over 20 epochs, spanning 4 days, ensuring that it converged adequately and learned effectively from the training data.

**Datasets:** We evaluated MOTE on three major datasets: MOT17 (Milan et al., 2016), MOT20 (Dendorfer et al., 2020), DanceTrack (Sun et al., 2022) and SportsMOT (Cui et al., 2023). MOT17 consists of 7 training and 7 testing sequences, primarily featuring crowded street scenes. MOT20, with 4 training and 4 testing sequences, offers more crowded scenes with higher levels of occlusion. DanceTrack contains 100 sequences, focusing on scenarios with high inter-object similarity and complex motion patterns.

**Metrics:** The evaluation followed standard MOT protocols, using metrics such as higher order tracking accuracy (HOTA) (Luiten et al., 2021), association accuracy (AssA), detection accuracy (DetA), ID F1 score (IDF1) (Ristani et al., 2016), multi-object tracking accuracy (MOTA) (Bernardin & Stiefelhagen, 2008), and ID switches (IDS).

## 4.2. State-of-the-Art Comparison

We compared MOTE against both CNN-based and Transformer-based methods on MOT17 (Milan et al., 2016), MOT20 (Dendorfer et al., 2020), DanceTrack (Sun et al., 2022). On MOT17 (Milan et al., 2016), as shown in Table 1, MOTE achieved the highest scores across all key metrics, including HOTA, AssA, DetA, IDF1, and MOTA, while

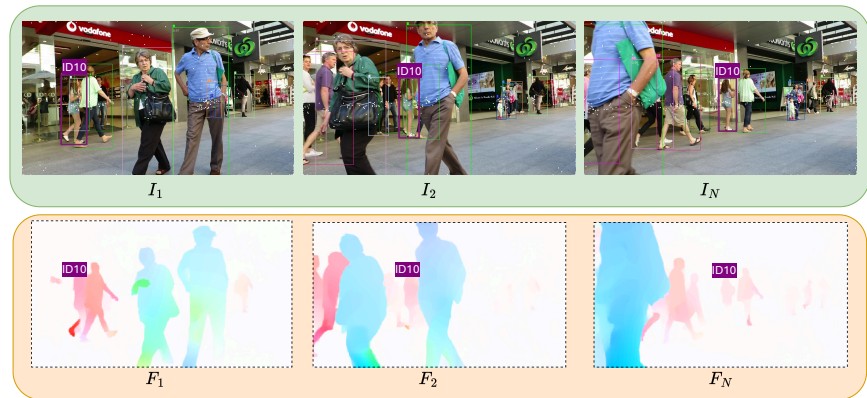

*Figure 5.* Illustration of MOTE's capability to track occluded subjects in challenging scenarios using optical flow. The top row shows consecutive frames $(I_1, I_2, \ldots, I_N)$ from a video sequence, where multiple subjects, including ID10, are partially occluded. The bottom row displays the corresponding flow fields $(F_1, F_2, \ldots, F_N)$ computed by the optical flow estimator. These flow fields capture the motion between consecutive frames, allowing our method to maintain accurate tracking of occluded subjects by leveraging the motion information. The effectiveness of our approach in handling occlusions is evident from the continuous tracking of ID10 across frames, even when it becomes partially occluded by other subjects. Note how the flow field maintains strong motion cues (shown in color intensity) even in regions of partial occlusion, enabling MOTE to predict object trajectories through occlusion events. The flow visualization uses standard color coding: hue indicates direction and saturation indicates magnitude of motion.

also reducing identity switches compared to other methods. Specifically, MOTE outperforms the second-best method, StrongSORT (Du et al., 2023), in HOTA by 2.8 points (66.3 vs. 63.5) and the second-best method, ByteTrack (Zhang et al., 2022b), in MOTA by 1.7 points (82.0 vs. 80.3). Furthermore, MOTE reduces identity switches by 2.4% (1412 vs. 1446) compared to StrongSORT, emphasizing its robustness in handling complex tracking scenarios with frequent occlusions. Table 2 demonstrates MOTE's superior performance on the MOT20 (Dendorfer et al., 2020)dataset, where it achieves the highest scores in HOTA, AssA, DetA, IDF1, and MOTA. This dataset provides a more challenging environment with dense crowds and severe occlusions. Thus, Table 2 confirms MOTE's effectiveness in tracking in crowded and occlusion-heavy environments. The DanceTrack (Sun et al., 2022) dataset, known for its high inter-object similarity and complex motion patterns, further tests the robustness of tracking methods. As shown in Table 6, MOTE achieves the top performance in HOTA, AssA, and MOTA, surpassing other methods. These results indicate MOTE's capability in challenging tracking tasks that involve intricate movements and interactions. Furthermore, to demonstrate cross-dataset generalization, we evaluated MOTE on MOT15 without fine-tuning, achieving 57.2 HOTA compared to MOTR's 28.4 (see Table 4 in Appendix), confirming our method's robustness across different annotation protocols and video conditions.

### 4.3. Preliminary Results on SportsMOT

To demonstrate MOTE's generalization capabilities beyond pedestrian tracking, we conducted preliminary experiments

*Table 1.* Comparative performance evaluation on the MOT17 (Milan et al., 2016) dataset, highlighting the best methods in CNN-based and Transformer-based categories. Metrics such as HOTA, AssA, DetA, MOTA, and IDF1 are considered. The best results for each metric are highlighted in bold, with the second-best shown in blue. The symbol / indicates unreported values.

| Methods | HOTA↑ | AssA↑ | DetA↑ | MOTA↑ | IDF1↑ | IDS↓ |
|---|---|---|---|---|---|---|
| *CNN-based:* | | | | | | |
| Tracktor++(Bergmann et al., 2019) | 44.8 | 45.1 | 44.9 | 53.5 | 52.3 | 2072 |
| CenterTrack(Zhou et al., 2020) | 52.2 | 51.0 | 53.8 | 67.8 | 64.7 | 3039 |
| TraDeS (Pang et al., 2021) | 52.7 | 50.8 | 55.2 | 69.1 | 63.9 | 3555 |
| QDTrack (Pang et al., 2021) | 53.9 | 52.7 | 55.6 | 68.7 | 66.3 | 3378 |
| GSDT (Wang et al., 2021c) | 55.5 | 54.8 | 56.4 | 66.2 | 68.7 | 3318 |
| FairMOT(Zhang et al., 2021) | 59.3 | 58.0 | 60.9 | 73.7 | 72.3 | 3303 |
| CorrTracker (Wang et al., 2021a) | 60.7 | 58.9 | 62.9 | 76.5 | 73.6 | 3369 |
| GRTU (Wang et al., 2021b) | 62.0 | 62.1 | 62.1 | 74.9 | 75.0 | 1812 |
| MAATrack (Stadler & Beyerer, 2022) | 62.0 | 60.2 | 64.2 | 79.4 | 75.9 | 1452 |
| StrongSORT (Du et al., 2023) | 63.5 | 63.7 | 63.6 | 78.3 | 78.5 | 1446 |
| ByteTrack (Zhang et al., 2022b) | 63.1 | 62.0 | 64.5 | 80.3 | 77.3 | 2196 |
| BoostTrack (Zhang et al., 2023a) | 65.4 | 64.2 | 64.8 | 80.5 | 80.2 | 1104 |
| *Transformer-based:* | | | | | | |
| TrackFormer (Meinhardt et al., 2021) | / | / | / | 65.0 | 63.9 | 3528 |
| TransTrack(Sun et al., 2020) | 54.1 | 47.9 | 61.6 | 74.5 | 63.9 | 3663 |
| MOTR(Zeng et al., 2022) | 57.8 | 55.7 | 60.3 | 73.4 | 68.6 | 2439 |
| MOTRv2(Zhang et al., 2023b) | 62.0 | 60.6 | 63.8 | 78.6 | 75.0 | / |
| **MOTE (Ours)** | **66.3** | **67.8** | **65.4** | **82.0** | **80.3** | **1412** |

on the SportsMOT dataset without retraining. We evaluated our method on three sequences and compared against Byte-Track and MOTR under identical conditions. As shown in Table 3, MOTE achieves the highest MOTA score at 45.7%, significantly outperforming ByteTrack (17.9%) and slightly surpassing MOTR (44.1%). Similarly, MOTE achieves an IDF1 score of 50.2%, compared to 31.4% for ByteTrack and 48.7% for MOTR. These results highlight MOTE's superior tracking accuracy and adaptability, even when applied to a new domain without additional training. While fine-tuning on SportsMOT could further enhance performance, these zero-shot results already demonstrate the robustness of our

approach in diverse tracking scenarios involving athletic movements and rapid motion changes.

*Table 2.* Comprehensive performance comparison on the MOT20 (Dendorfer et al., 2020) dataset. The highest scores are highlighted in bold and the second-best scores are in blue. The symbol / indicates unreported values.

| Methods | HOTA↑ | AssA↑ | DetA↑ | MOTA↑ | IDF1↑ |
|---|---|---|---|---|---|
| FairMOT (Zhang et al., 2021) | 54.6 | 54.7 | 54.7 | 61.8 | 67.3 |
| ByteTrack (Zhang et al., 2022b) | 61.3 | 59.6 | 62.9 | 76.2 | 75.2 |
| OC-SORT (Cao et al., 2022) | 62.4 | 62.5 | / | 75.9 | 76.4 |
| MOTRv2 (Zhang et al., 2023b) | 60.3 | 58.1 | 62.9 | 76.2 | 72.2 |
| StrongSORT (Du et al., 2023) | 61.5 | 62.5 | 59.9 | 72.2 | 75.9 |
| BoostTrack (Zhang et al., 2023a) | 63.0 | 62.8 | 63.4 | 76.4 | 76.5 |
| **MOTE (Ours)** | **65.8** | **66.9** | **64.9** | **81.7** | **79.8** |

*Table 3.* Zero-shot evaluation on SportsMOT (Cui et al., 2023) dataset (3 sequences). All methods were evaluated without retraining.

| Methods | MOTA↑ | IDF1↑ | FPS↑ |
|---|---|---|---|
| ByteTrack (Zhang et al., 2022b) | 17.9 | 31.4 | 30.2 |
| MOTR (Zeng et al., 2022) | 44.1 | 48.7 | 7.5 |
| **MOTE (Ours)** | **45.7** | **50.2** | **22.2** |

*Table 4.* Extended results comparing MOTR and MOTE on the MOT15 dataset, with models trained on the MOT17 dataset.

| Methods | HOTA↑ | MOTA↑ | IDF1↑ |
|---|---|---|---|
| MOTR | 28.4 | 32.5 | 36.3 |
| MOTE (Ours) | **57.2** | **63.2** | **68.8** |

## 4.4. Computational Analysis

MOTE's computational overhead is carefully optimized to balance performance gains with practical deployment considerations. Table 5 presents a detailed breakdown of processing times for each component. Our framework adds only 25ms per frame compared to MOTR's baseline of 133ms (7.5 FPS) on high-resolution inputs (1536×800), representing a 19% increase in computation time. This modest overhead stems from RAFT optical flow estimation (18ms with 20 iterations) and the softmax splatting layer (7ms).

For real-time applications, we explored lightweight alternatives: using fewer RAFT iterations (10 instead of 20) reduces overhead to 12ms with only 1.8% HOTA reduction. Additionally, mixed-precision training and inference provide 35% speedup with negligible accuracy loss.

## 4.5. Occlusion Handling and Qualitative Analysis

MOTE's ability to maintain accurate tracking under prolonged occlusion conditions is one of its key strengths. Figure 5 illustrates MOTE's effectiveness in tracking subjects through both partial and complete occlusions that persist over multiple frames. The top row shows consecutive frames from a video sequence, where multiple subjects, including

*Table 5.* Computational breakdown on 1536×800 resolution (A100 GPU).

| Component | Time (ms) | % of Total |
|---|---|---|
| Deformable DETR Encoder | 45 | 28.1% |
| RAFT (20 iterations) | 18 | 11.3% |
| Softmax Splatting | 7 | 4.4% |
| ETEM | 12 | 7.5% |
| Decoder | 48 | 30.0% |
| Other Components | 30 | 18.7% |
| **Total** | **160** | **100%** |

ID10, experience significant occlusions. The bottom row displays the corresponding optical flow fields computed by our method. These flow fields capture the motion between consecutive frames, allowing MOTE to maintain accurate tracking of occluded subjects by leveraging motion information, even when visual evidence is limited. The continuous tracking of ID10 across frames, even during prolonged occlusions, demonstrates MOTE's robustness in handling one of the most challenging aspects of tracking in crowded environments. For detailed analysis of extended occlusion scenarios, we refer readers to Section **??** in the Appendix.

*Table 6.* Performance evaluation on the DanceTrack (Sun et al., 2022) dataset comparing CNN-based and Transformer-based methods. The best-performing scores are shown in bold, while the second-best are highlighted in blue. MOTRv2* denotes MOTRv2 with an extra association, adding validation set for training, and test ensemble.

| Methods | HOTA↑ | AssA↑ | DetA↑ | MOTA↑ | IDF1↑ |
|---|---|---|---|---|---|
| CNN-based: | | | | | |
| FairMOT (Zhang et al., 2021) | 39.7 | 23.8 | 66.7 | 82.2 | 40.8 |
| CenterTrack (Zhou et al., 2020) | 41.8 | 22.6 | 78.1 | 86.8 | 35.7 |
| TraDeS (Pang et al., 2021) | 43.3 | 25.4 | 74.5 | 86.2 | 41.2 |
| QDTrack (Pang et al., 2021) | 54.2 | 38.7 | 81.0 | 87.7 | 50.4 |
| ByteTrack (Zhang et al., 2022b) | 47.7 | 31.0 | 71.0 | 91.5 | 48.8 |
| OC-SORT (Cao et al., 2022) | 55.1 | 38.3 | 80.3 | 92.0 | 54.6 |
| Transformer-based: | | | | | |
| TransTrack (Sun et al., 2020) | 45.5 | 27.5 | 75.9 | 88.4 | 45.2 |
| GTR (Wang et al., 2021b) | 48.0 | 31.9 | 72.5 | 89.7 | 50.3 |
| MOTRv2 (Zhang et al., 2023b) | 69.9 | 59.0 | 83.0 | 91.9 | 71.7 |
| MOTRv2* (Zhang et al., 2023b) | 73.4 | 64.4 | **83.7** | 92.1 | **76.0** |
| **MOTE (Ours)** | **74.2** | **65.2** | 82.6 | **93.2** | 75.2 |

## 4.6. Ablation Study

We conducted an ablation study to assess the impact of different components and parameters within MOTE. The study focused on three main aspects: the choice of splatting technique, the number of iterations in optical flow estimation, and the effect of occlusion weights. To enable rapid experimentation and fair component comparison, all ablation models were trained for 5 epochs, providing clear insights into the relative importance of each component while maintaining reasonable training times.

As presented in Table 7, softmax splatting outperforms

linear splatting across all metrics, including a 3.2-point increase in HOTA, a 3.6-point increase in MOTA, and a 3.5-point increase in IDF1, along with a reduction of 316 identity switches. These significant improvements, achieved even with limited training, highlight the fundamental advantage of the softmax splatting approach.

We compared linear splatting and softmax splatting on the MOT17 dataset. As presented in Table 7, softmax splatting outperforms linear splatting across all metrics, including a 3.2-point increase in HOTA, a 3.6-point increase in MOTA, and a 3.5-point increase in IDF1, along with a reduction of 316 identity switches. These findings highlight the effectiveness of Softmax Splatting in enhancing tracking accuracy.

We also examined the effect of varying the 'iters' parameter in the optical flow estimation process. Table 8 indicates that 20 iterations offer the best balance between tracking performance and computational complexity, achieving the highest HOTA and IDF1 scores. While 25 iterations slightly improve MOTA, the gains in other metrics diminish, suggesting that 20 iterations provide optimal performance.

**Effect of Occlusion Weights:** Lastly, we evaluated the impact of incorporating occlusion weights into MOTE's loss function. As shown in Table 9, the inclusion of occlusion weights results in a significant performance improvement, with increases of 2.8 points in HOTA, 1.7 points in MOTA, and a reduction of 34 identity switches compared to the configuration without occlusion weights.

*Table 7.* Ablation study comparing linear splatting and softmax splatting on the MOT17 dataset. The models were trained for 5 epochs.

| Method | HOTA↑ | MOTA↑ | IDF1↑ | IDS↓ |
|---|---|---|---|---|
| Linear Splatting | 55.2 | 61.3 | 65.7 | 2450 |
| Softmax Splatting | **58.4** | **64.9** | **69.2** | **2134** |

*Table 8.* Ablation study for the 'iters' parameter in the forward method of optical flow estimation on the MOT17 dataset.

| iters | HOTA↑ | MOTA↑ | IDF1↑ | IDS↓ |
|---|---|---|---|---|
| 15 | 56.1 | 62.4 | 66.8 | 2300 |
| 20 | **58.3** | 63.7 | **69.0** | 2205 |
| 25 | 57.4 | **64.5** | 68.1 | **2150** |

## 5. Conclusion

In this paper, we introduced MOTE, a novel approach to multi-object tracking that excels in complex scenarios with frequent occlusions. By combining softmax splatting, deformable DETR, and optical flow estimation, MOTE consis-

*Table 9.* Ablation study on MOT17 (Milan et al., 2016) comparing loss functions with and without occlusion weights.

| Config. | HOTA↑ | MOTA↑ | IDF1↑ | IDS↓ |
|---|---|---|---|---|
| Without Occ. weights | 63.5 | 80.3 | 78.5 | 1446 |
| With Occ. weights | **66.3** | **82.0** | **80.3** | **1412** |
| Improvement | +2.8 | +1.7 | +1.8 | -34 |

tently maintains accurate and reliable object tracking, even under challenging occlusion conditions. MOTE demonstrates superior performance across MOT17, MOT20, and DanceTrack datasets, setting new state-of-the-art benchmarks. The results clearly demonstrate MOTE's superior performance, surpassing current techniques in the field. While highly effective, our method faces challenges with small or distant objects, which are inherent difficulties of optical flow-based methods. Future work will focus on addressing these limitations through alternative motion estimation methods and the integration of additional sensor modalities such as depth cameras or LiDAR. We also see significant potential in leveraging graph neural networks to enhance the detection and tracking of challenging objects, making tracking systems even more reliable and effective in diverse and demanding environments. Looking forward, we plan to extend MOTE's capabilities in several directions. First, we will explore transformer-based optical flow methods that could be trained end-to-end with our tracking framework. Second, we aim to incorporate long-term feature banks for handling extended occlusions beyond our current temporal window. Third, we plan comprehensive evaluation on additional challenging datasets including KITTI for autonomous driving scenarios and BFT for non-human object tracking. Finally, we are investigating efficient deployment strategies for edge devices, including model quantization and architecture search tailored for real-time applications.

## Impact Statement

This work advances multi-object tracking technology with implications for various societal applications. While MOTE can enhance public safety through improved surveillance systems and enable safer autonomous navigation, we acknowledge potential privacy concerns. The ability to track individuals through occlusions could be misused for unauthorized surveillance. We advocate for responsible deployment with appropriate privacy safeguards and transparent policies. Additionally, while our method is computationally efficient compared to its performance gains, the environmental impact of training and deploying such models should be considered.

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
