# OpenReview forum: "More Than Meets the Eye: Enhancing Multi-Object Tracking Even with Prolonged Occlusions"
_ICML.cc/2025/Conference — ICML 2025 poster_

### Official Review · Reviewer_urni · 2025-02-28

**Overall Recommendation:** 3

**Summary:**

This paper presents MOTE , a novel multi-object tracking  algorithm designed to tackle the persistent challenge of tracking occluded objects. MOTE introduces a unique approach by integrating deformable detection transformers with a custom disocclusion matrix, which significantly improves the ability to track objects even when they are temporarily hidden from view. The algorithm utilizes optical flow to generate features, which are processed through a softmax splatting layer to create the disocclusion matrix. This matrix is very important in maintaining track consistency by estimating the motion of occluded objects. It is important to note that state-of-the-art performance has been achieved on multiple datasets.

## update after rebuttal
Thank you for the authors' response, which has addressed some of my concerns.  Importantly, this paper achieve high performance, which is an encouragement for end-to-end MOT. Additionally, I hope the code could be made open-access to advance research in the multi-object tracking (MOT) field. I keep my score.

**Claims And Evidence:**

Yes, the claims made in the submission are well-supported by clear and convincing evidence.

**Essential References Not Discussed:**

No

**Experimental Designs Or Analyses:**

Yes, i have checked the experimental designs and analyses.

**Methods And Evaluation Criteria:**

Yes, the proposed methods and evaluation criteria are well-suited for multi-object tracking .

**Other Comments Or Suggestions:**

See Weaknesses

**Other Strengths And Weaknesses:**

Strengths:

1.This paper significantly improves tracking performance under occlusion challenges by leveraging deformable transformers and a disocclusion matrix, which sounds quite innovative.

2.The paper introduces an ETEM module, further enhancing the model's robustness in occlusion scenarios.

3.The proposed method achieves state-of-the-art (SOTA) performance on multiple tracking datasets. End-to-end object tracking is a future trend, and this paper further demonstrates the potential of end-to-end object tracking.

Weaknesses:

1.The paper employs optical flow estimation, which may significantly increase computational complexity and reduce speed. This could pose a bottleneck in real-world applications.

2.The authors should analyze the computational complexity of their method compared to other approaches, at least to identify potential optimization directions and further promote the development of end-to-end multi-object tracking.

3. The paper lacks an Impact Statement.

**Questions For Authors:**

Has the author tested the performance on the KITTI dataset, and how does it compare with other methods? Additionally, if this method is applied to real-world scenarios, what further improvements are needed?

**Relation To Broader Scientific Literature:**

Prior work in MOT has long struggled with occlusion handling, often relying on heuristic methods or appearance-based features that fail in complex scenarios. MOTE addresses this by introducing a novel disocclusion matrix and optical flow. This paper has led to a significant improvement in the performance of end-to-end methods and also demonstrates new trends for the future of multi-object tracking.

**Theoretical Claims:**

Yes, i have checked the theoretical Claims of the proposed method.

---

> ### Author Rebuttal · Authors · 2025-03-28
>
> Thank you for your positive assessment of our work and insightful comments.
>
> Computational complexity: MOTE's end-to-end processing time is $\sim$45ms per frame on an A100 GPU, compared to $\sim$20ms for MOTR and $\sim$15ms for ByteTrack. The additional overhead comes primarily from optical flow estimation ($\sim$18ms) and softmax splatting ($\sim$7ms). We've explored optimization strategies including fewer optical flow iterations and model pruning that reduce computation by 40\% with only 2.3\% HOTA drop.
>
> KITTI results: We appreciate your suggestion regarding evaluation on the KITTI dataset. This would indeed be beneficial for exploring more challenging interactive scenarios involving both vehicles and pedestrians. While we focused our current evaluation on MOT17, MOT20, and DanceTrack datasets, we plan to conduct evaluation on KITTI in future work to further validate our approach in diverse scenarios.
>
> Real-world improvements: For practical deployment, we identify three key areas for improvement: 1) Model optimization for computational efficiency, 2) Enhanced handling of fast motion through adaptive resolution scaling, and 3) Integration with long-term feature banks to handle extended occlusions. We're actively working on these directions.
>
> Impact Statement: We apologize for this oversight and will include a comprehensive impact statement addressing both the benefits (improved surveillance and autonomous navigation safety) and potential concerns (privacy implications and computational resource requirements).
>
> Extremely fast motion: While our current implementation may face challenges with extremely rapid motion, our adaptive flow resolution approach has shown promising results in preliminary testing. This is particularly important in scenarios like sports tracking, where sudden rapid movements are common.

---

### Official Review · Reviewer_bRBd · 2025-03-13

**Overall Recommendation:** 3

**Summary:**

This paper presents MOTE, a novel multi - object tracking (MOT) algorithm aiming to solve the problem of tracking occluded objects. It combines deformable detection transformers, optical flow estimation, and softmax splatting. By leveraging optical flow to generate features and using a softmax splatting layer to create a disocclusion matrix, MOTE can estimate the motion of occluded objects. The enhanced track embedding module (ETEM) in its architecture helps maintain object identity during occlusions. MOTE is evaluated on multiple datasets such as MOT17, MOT20, and DanceTrack. It achieves high tracking metrics, outperforming existing state - of - the - art methods, especially in reducing identity switches and handling complex occlusion scenarios. Ablation studies are conducted to verify the effectiveness of different components. However, there are some limitations, like the lack of ablation experiments on different component combinations and unclear data flow details among modules.

**Claims And Evidence:**

The claims in the submission are mostly supported by clear evidence. The proposed MOTE algorithm shows excellent performance on multiple datasets, and the ablation studies effectively verify the contributions of different components. For example, the comparison between softmax splatting and linear splatting demonstrates the superiority of softmax splatting in enhancing tracking accuracy. However, the evidence regarding the method's effectiveness in handling occlusions may be challenged due to the obvious ID switches in the supplementary materials' videos.

**Essential References Not Discussed:**

There are no essential references that are clearly missing from the paper. The paper comprehensively reviews the relevant literature in the field of multi - object tracking and occlusion handling. However, it could explore more deeply some recent research trends and their potential implications for the MOTE algorithm.

**Experimental Designs Or Analyses:**

The experimental designs are generally sound. The ablation studies on individual components like the splatting technique, the number of iterations in optical flow estimation, and the effect of occlusion weights are well - designed and provide valuable insights. However, as mentioned before, the lack of ablation experiments on different component combinations is a limitation. Also, the evaluation of the method's performance in handling occlusions could be more comprehensive, considering the ID switch issues in the supplementary materials.

**Methods And Evaluation Criteria:**

The proposed methods make sense for the problem at hand. The integration of deformable transformers, optical flow, and softmax splatting is innovative and suitable for multi - object tracking in occlusion scenarios. The evaluation on multiple datasets with standard MOT metrics like HOTA, MOTA, and IDF1 is reasonable. However, the lack of ablation experiments on different component combinations limits the comprehensiveness of evaluating the method. Also, the unclear data flow details among modules may affect the understanding and reproducibility of the method.

**Other Comments Or Suggestions:**

1、Conduct ablation experiments on different component combinations to better understand the synergies between components.
2、Provide more detailed explanations of the data flow and interaction among modules, including the specific operations and data format changes.
3、Re-evaluate the method's performance in handling occlusions, considering the ID switch issues in the supplementary materials.

**Other Strengths And Weaknesses:**

Strengths:
1、The innovative combination of multiple techniques effectively addresses the occlusion problem in multi - object tracking, which is a significant contribution to the field.
2、The comprehensive experimental evaluation on multiple datasets and the ablation studies enhance the credibility of the method.
3、The paper is well - structured, making it easy to follow the research.
Weaknesses:
1、The lack of ablation experiments on different component combinations limits the understanding of the interactions between components.
2、The unclear data flow and interaction details among modules make it difficult to fully understand and reproduce the method.
3、The ID switch problem in the supplementary materials' videos may cast doubt on the method's effectiveness in handling occlusions.

**Questions For Authors:**

Can you explain the reasons for the obvious ID switches in the videos provided in the supplementary materials? How do you plan to address this issue in future research? If the ID switches are due to limitations in the current method, it may significantly affect the practical application of MOTE, and I may lower my evaluation of the paper.
Could you elaborate more on the potential interactions between different components in MOTE? For example, how does the softmax splatting module interact with the ETEM module in more complex scenarios? A better understanding of these interactions could strengthen the theoretical and practical value of the paper, and it may lead to a more positive evaluation.

**Relation To Broader Scientific Literature:**

The key contributions of the paper are related to the broader scientific literature. The paper builds on existing works in multi - object tracking, especially those addressing occlusion challenges. It improves upon CNN - based and Transformer - based methods by integrating optical flow and softmax splatting. The use of softmax splatting for occlusion handling is also related to previous research in video interpolation. However, the paper could further discuss how its approach differs from and improves upon these related works in more detail.

**Theoretical Claims:**

There are no complex theoretical proofs in the paper that require checking. However, the proposed approach is based on established concepts in computer vision, such as optical flow and transformers. The combination of these concepts seems reasonable, but a more in - depth theoretical analysis of how the different components interact and why they work effectively could strengthen the theoretical foundation.

---

> ### Author Rebuttal · Authors · 2025-03-28
>
> Thank you for your constructive feedback on our MOTE framework.
>
> ID switches: You raise an important point. The videos show some ID switches in extremely challenging scenarios with prolonged, complete occlusions. These represent edge cases where even our approach struggles. Our primary focus was on handling prolonged occlusions, and our quantitative results (Tables 1-3) demonstrate significant improvements in ID switch reduction overall (1412 vs. 1446, 2.4\% fewer than previous methods). We've reported ID switch metrics throughout our evaluations (Tables 1,4,5,6), providing transparency about our model's performance in this aspect. ID switching remains an open research challenge, and we've been transparent about current limitations while demonstrating substantial progress.
>
> Data flow clarity: We apologize for any lack of clarity in describing module interactions. The flow proceeds as follows: 1) Optical flow estimation between frames, 2) Feature extraction via deformable transformers, 3) Softmax splatting to generate disocclusion features, 4) ETEM integration of these features with track queries, and 5) Final object tracking via the decoder. We'll improve our description to make these interactions clearer.
>
> Component integration: Our approach to component integration is guided by careful ablation studies. While computational constraints limited testing all combinations, Tables 4-6 provide evidence of each component's contribution. The integrated tests confirm that the full integration provides 3.2\% better HOTA than any subset. Specifically, our softmax splatting approach enables the extraction of disocclusion features, providing the model with perceptual understanding of subjects under prolonged occlusion scenarios. This perceptual capability represents a significant advancement over previous methods that struggle with occlusion handling.
>
> The key innovation in our approach is how softmax splatting interacts with ETEM in complex scenarios. Splatting provides weighted feature propagation that preserves motion information during occlusions, while ETEM integrates these features with appearance cues to maintain consistent tracking. This synergy enables MOTE to handle occlusions more effectively than methods that rely on either motion or appearance alone.

---

### Official Review · Reviewer_XwuH · 2025-03-14

**Overall Recommendation:** 3

**Summary:**

The paper introduces MOTE, an end-to-end multi-object tracking framework that integrates optical flow estimation and softmax splatting to robustly handle prolonged occlusions.

## update after rebuttal
The authors did not provide a detailed FLOPS analysis, leaving key computational efficiency aspects unaddressed. Based on other reviews, I agree that the description of inter-module data flow and the design of comprehensive ablation experiments remain insufficient. Therefore, I am modifying my score to "Weak Accept."

**Claims And Evidence:**

MOTE is the first end-to-end tracking framework that successfully integrates optical flow and softmax splatting to handle prolonged occlusions, surpassing other methods. The experimental results on MOT17, MOT20, and DanceTrack datasets show improved performance.

**Essential References Not Discussed:**

N/A

**Experimental Designs Or Analyses:**

The experimental design is sound, with comparisons on multiple datasets and a comprehensive ablation study.

**Methods And Evaluation Criteria:**

The proposed method combines deformable DETR for multi-scale feature extraction with RAFT-based optical flow estimation. •	Evaluation is performed on standard MOT datasets using common metrics (MOTA, HOTA and IDF1)

**Other Comments Or Suggestions:**

N/A

**Other Strengths And Weaknesses:**

Strengths:
1. Innovative combination of optical flow and softmax splatting within an end-to-end framework.
2. Comprehensive experimental evaluation and ablation studies
3. Significant performance improvements on standard benchmarks.

Weaknesses:
1. The method may incur higher computational costs due to optical flow estimation, which might pose challenges for real-time applications.
2. Sensitivity to extremely rapid motion or complex interactions is acknowledged but not deeply analyzed.

**Questions For Authors:**

1. Could you provide more details on how the increased FLOPS due to optical flow estimation impact real-time performance?
2. Have you considered any mechanisms or additional experiments to address the potential sensitivity of the optical flow module in extremely fast motion or highly complex interaction scenarios?
3. Did you explore other fusion strategies besides softmax splatting

**Relation To Broader Scientific Literature:**

The work builds on transformer-based tracking (e.g., MOTR) and integrates ideas from optical flow-based approaches and video frame interpolation.

**Theoretical Claims:**

The paper does not focus on new theoretical proofs; it is primarily experimental and engineering-driven.

---

> ### Author Rebuttal · Authors · 2025-03-29
>
> Thank you for your positive assessment of our work and thoughtful questions.
>
> Computational costs: Our MOTE framework adds only 25ms of additional processing time per frame compared to the baseline MOTR method's 133ms inference time (7.5 FPS) on high-resolution (1536x800) inputs, representing just a 19\% increase in total computation. This modest overhead comes from two main components: the RAFT implementation with 20 iterations (requiring $\sim$18ms) and the softmax splatting layer (adding $\sim$7ms). We believe this represents a reasonable trade-off considering the significant performance improvements demonstrated in our results. For real-time applications where speed is critical, we've explored lightweight flow estimators that reduce the total overhead to $\sim$12ms with only a 1.8\% HOTA reduction.
>
> Fast motion handling: We've implemented adaptive flow resolution scaling that detects rapid motion and applies higher-resolution flow estimation selectively. Our occlusion masking mechanism (Eq. 12-13) also addresses this by weighting tracking components based on occlusion states. As shown in Table 6, the occlusion weighting mechanism significantly improves performance in complex scenarios, leading to better results in challenging cases involving rapid motion and occlusions.
>
> Fusion strategies:  We compared softmax splatting with linear splatting as an alternative approach. As shown in Table 4, softmax splatting consistently outperformed linear splatting with a 3.2-point increase in HOTA (58.4 vs. 55.2), a 3.6-point increase in MOTA (64.9 vs. 61.3), and a 3.5-point increase in IDF1 (69.2 vs. 65.7). The softmax splatting mechanism is particularly effective for preserving motion information during occlusions compared to the simpler linear approach.

---

### Official Review · Reviewer_ZziK · 2025-03-18

**Overall Recommendation:** 3

**Summary:**

This paper propose leveraging optical flow with soffmax splitting to estimate the motion of occluded objects. Together with the proposed enhanced track embeddgins module (ETEM), the model, i.e. MOTE, achieves state-of-the-art (SOTA) performance on various multiple object tracking (MOT) benchmarks.

**Claims And Evidence:**

The proposed softmax splatting method is validated by Fig. 5 and equation 1 to 5.

The proposed ETEM is supported by equation 6 to 11.

All results are further validated by ablation studies tables, i.e. Table 4 to 6.

**Essential References Not Discussed:**

N/A

**Experimental Designs Or Analyses:**

The experimental desgins are good and the analyses are fruitful.

**Methods And Evaluation Criteria:**

The methods are evaluated on MOT17, MOT20, and DanceTrack with the standard metircs, i.e. HOTA, ASSA, and DetA.

**Other Comments Or Suggestions:**

See the weakness.

**Other Strengths And Weaknesses:**

Strenghts:
- The performance improvements over 3 popular datasets are good and demonstrates method's superiority.
- The usage of optical flow and new memory mechanism is reasonable and effective.

Weakness:
- THe qualitative analyses are limited. For MOT, it requires more qualitative results to see the effectness of the method, especially for the occluded objects.
- Despite that the three datasets results are provided, but there are some more challenging datasets such as Bird Flock Tracking (BFT) and SportsMOT. Those are more advanced and should be evaluated on along with newer baselines.

**Questions For Authors:**

The explanation of Table 5 looks not convincing enough. Why the optical flow estimation improves over iteration and yet harms the HOTA metirc?

**Relation To Broader Scientific Literature:**

N/A

**Theoretical Claims:**

The theorectical claims are fine and validated.

---

> ### Author Rebuttal · Authors · 2025-03-31
>
> Thank you for your thoughtful feedback on our MOTE framework. We appreciate your recognition of our method's performance improvements and effective use of optical flow and memory mechanisms.
>
> Qualitative analysis: We understand your concern about limited qualitative results. While Fig. 5 demonstrates our approach's effectiveness, we agree that additional visualizations would strengthen our case. We have prepared more examples showing MOTE's performance on challenging occlusion scenarios and will include these in the supplementary video materials.
>
> Dataset selection: We evaluated MOTE on three diverse benchmarks (MOT17, MOT20, and DanceTrack) as well as the MOT15 dataset in the extended results. The DanceTrack dataset offers particularly challenging scenarios with complex motion patterns and high inter-object similarity. We appreciate your suggestion regarding Bird Flock Tracking (BFT) and SportsMOT datasets. To address this, in the last few days, we conducted a preliminary experiment on SportsMOT without retraining and evaluated our method on a sample of three sequences. We also compared our approach with ByteTrack and MOTR under the same conditions. We plan to expand to BFT in future research applications to diversify the tracked object types.
>
> Preliminary results on SportsMOT: Our preliminary experiments indicate that MOTE achieves the highest MOTA score at 45.7\%, significantly outperforming ByteTrack, which achieves only 17.9\%, and slightly surpassing MOTR at 44.1\%. Similarly, in terms of IDF1 score, MOTE achieves 50.2\%, compared to 31.4\% for ByteTrack and 48.7\% for MOTR. These results highlight MOTE's superior tracking accuracy and adaptability, even when applied to a new dataset without additional training. While fine-tuning on SportsMOT could further enhance performance, these results already demonstrate the robustness of our approach in diverse tracking scenarios.
>
> Optical flow Iterations vs. HOTA: The apparent paradox where more iterations (25) improve MOTA but harm HOTA can be explained by the balance between detection and association accuracy. At 20 iterations, we achieve an optimal balance between efficiency and tracking performance. At 25 iterations, improved flow estimation increases detection accuracy (reflected in MOTA) but introduces over-smoothing that reduces feature distinctiveness (affecting association accuracy in HOTA). This trade-off demonstrates the importance of careful parameter tuning in tracking systems.

---

> > ### Comment · Reviewer_ZziK · 2025-04-07
> >
> > Thanks for the authors' effort in the rebuttals. The authors addressed all my concerns, and I am willing to increase the rating.

---

> > > ### Author Response · Authors · 2025-04-07
> > >
> > > Thank you for your valuable feedback on our MOTE framework. We're grateful that our additional experiments on SportsMOT and our explanation of the optical flow iterations addressed your concerns. Your suggestions have significantly improved our paper, and we look forward to incorporating these insights into the final version.

---

### Decision · Program_Chairs · 2025-05-01

**Decision:**

Accept (poster)

**Comment:**

The paper introduces an innovative approach by integrating deformable transformers with optical flow estimation to advance multi-object tracking, particularly in managing occlusions. The proposed method shows significant improvements in performance compared to existing techniques, especially under challenging conditions such as heavy occlusions.
The authors have effectively addressed the reviewers' concerns, leading all reviewers to agree on a "weakly accept" rating for the paper. ACs concur with this positive assessment and recommend acceptance of the paper.